# A Calculation Method of Passenger Flow Distribution in Large-Scale Subway Network Based on Passenger–Train Matching Probability

**DOI:** 10.3390/e24081026

**Published:** 2022-07-26

**Authors:** Guanghui Su, Bingfeng Si, Kun Zhi, He Li

**Affiliations:** 1School of Traffic and Transportation, Beijing Jiaotong University, Beijing 100044, China; 18114011@bjtu.edu.cn (G.S.); zhikun2017@126.com (K.Z.); 2Beijing Metro Network Administration Co., Ltd., Beijing 100101, China; lihe@bmncc.com.cn

**Keywords:** subway network, passenger flow distribution, data driven, passenger–train matching, time-dependent

## Abstract

The ever-increasing travel demand has brought great challenges to the organization, operation, and management of the subway system. An accurate estimation of passenger flow distribution can help subway operators design corresponding operation plans and strategies scientifically. Although some literature has studied the problem of passenger flow distribution by analyzing the passengers’ path choice behaviors based on AFC (automated fare collection) data, few studies focus on the passenger flow distribution while considering the passenger–train matching probability, which is the key problem of passenger flow distribution. Specifically, the existing methods have not been applied to practical large-scale subway networks due to the computational complexity. To fill this research gap, this paper analyzes the relationship between passenger travel behavior and train operation in the space and time dimension and formulates the passenger–train matching probability by using multi-source data including AFC, train timetables, and network topology. Then, a reverse derivation method, which can reduce the scale of possible train combinations for passengers, is proposed to improve the computational efficiency. Simultaneously, an estimation method of passenger flow distribution is presented based on the passenger–train matching probability. Finally, two sets of experiments, including an accuracy verification experiment based on synthetic data and a comparison experiment based on real data from the Beijing subway, are conducted to verify the effectiveness of the proposed method. The calculation results show that the proposed method has a good accuracy and computational efficiency for a large-scale subway network.

## 1. Introduction

The subway system plays an increasingly important role in urban transportation due to its characteristics of reliability, punctuality, and high capacity. Taking Beijing as an example, in the past 10 years (2010–2019), the average annual growth rate of subway passenger trips in Beijing has reached 115.5%, and the sharing rate of subway has increased from 23% to 47.2% (Beijing Transportation Institute, 2019). The influx of passengers into the subway system causes the crowding of passengers on platforms and inside subway carriages, which not only negatively affects the passengers’ perception, but also challenges the safety and efficiency of subway train operation. With this concern, it is urgent for subway operators to accurately estimate the passenger flow distribution throughout the network such that operation strategies [1] and emergency plans [2] can be designed appropriately.

The traditional research for estimating passenger flow distribution can be divided into two categories: (1) simulation method and (2) mathematical model. The core idea of the former is to depict the passenger flow evolution in a subway network by simulating the passenger’s behavior [3,4,5], while the latter is to formulate an equivalent mathematical model by analyzing passenger route choice behavior based on travel cost [6,7,8]. Generally, these studies are based on the following assumptions: (1) each train has a fixed capacity [3,4,6,8]; (2) the passenger boarding process follows the FCFS (First-Come-First-Served) principle [6,7]; (3) ignoring the arrival time of an individual passenger [3,4,6,7,9]; and (4) neglecting the impact of network time-dependent state on passenger choice behavior [3,4,7]. However, the assumptions restrict the traditional methods in accurately depicting the factors influencing the passenger flow distribution [10,11,12], such as the in-train congestion [13], the passengers’ psychology during their travel process [14] and so on. On the other hand, these traditional methods cannot depict the impact of the time-dependent state of a subway network (such as passenger retention, train overload, etc.) because they focus on understanding the passenger flow distribution from the aggregated level, but not from the level of data and a disaggregated level.

The automatic fare collection (AFC) system has been widely used in subway systems and provides a data-driven approach for analyzing the passengers’ choice behavior [15] and the passenger flow distribution in a subway system [16,17]. For example, Zhang [18] estimated the network-wide link travel time and station waiting time using AFC data in an urban rail transit system. Chen [19] proposed a methodology to mine passenger travel patterns based on AFC data and automatic vehicle location data. Nevertheless, the passenger trajectories in the subway network are hard to be easily obtained because: (1) passenger travel behavior is affected by individual subjective factors; and (2) the subway network has strong time-dependent characteristics [20,21]. Accordingly, the problem of accurately calculating passenger flow distribution in a subway network based on multi-source data has attracted more and more attention. At the very beginning, Kusakabe et al. [22] enumerated all train combinations for passengers according to their tap-in and tap-out time and then inferred the train picked by the passenger according to the strategy of “minimum waiting time—minimum egress time—the least number of transfer”. Their work laid a foundation for the comprehensive use of AFC data and train timetables to estimate passenger train choice and flow distribution. Subsequently, many research studies related to passenger flow distribution have been developed based on AFC data and train timetables. For example, Zhu and Xu [23] proposed an individual-based passenger flow model by enumerating their path’s boarding plans; however, the model cannot be used in a large-scale network due to the limitation of computational efficiency. Zhao et al. [24] assumed that the walking time of access/egress/transfer is shorter than the headway and established a probability model to convert the problem of passenger route choice into the probability of taking different trains. However, their model cannot be applicable to a high-frequency subway network. Zhu et al. [25] proposed a probability-based passenger-to-train assignment model to infer the most likely train for passengers; however, their model cannot be directly extended to the network level because the factors influencing passenger’s behavior such as transfer and path choice are not considered. Hörcher et al. [13] extended the passenger-to-train model to the network level while considering the factor of in-train congestion. However, the transfer congestion that also impacts passenger route choice is not fully considered. In addition, the computation efficiency of the model is a relevant issue. More recently, Mo et al. [5] proposed a Bayesian optimization method based on AFC data to identify the optimal capacity constraints of trains and then used a timetable-based network loading model to calculate the passenger flow distribution. Zhu et al. [26] proposed an integrated probability model for calculating passenger path choice and itinerary.

Although existing studies have made considerable contributions to the estimation of passenger train choice or subway passenger flow distribution based on multi-source data, there are still many issues that need improvement. For example, ignoring or simplifying the important factors such as transfer, network time-dependent characteristics [13,25] and so on. Simultaneously, with the increase of passenger volume and travel distance, the data-driven model mentioned above will be very difficult for a large subway network due to its computational efficiency. In view of these unsolved issues, this paper proposes a probability model based on multi-source data (including AFC data, train timetables and network topology data) to estimate the passenger travel trajectory and then calculate the passenger flow temporal and spatial distribution throughout a subway network. In addition, a reverse derivation method, which can reduce the scale of possible train combinations for passengers, is proposed to improve the computational efficiency. The main contributions of this paper are specifically listed as follows:(1)A data-driven passenger–train matching probability model is proposed. In this model, the dynamic time–space trajectory of each individual passenger is explicitly characterized by mining AFC data, train timetables and network topology.(2)According to the consistency characteristics of passenger travel behavior [10,23] and the topology data of the subway network, a reverse derivation method is proposed to decompose the passenger itinerary network into multiple small subnets. This method can reduce the scale of passenger itineraries without affecting the accuracy of the model to avoid many unnecessary calculations and improve the computational efficiency. Therefore, the model can be applied to calculate the passenger flow distribution in a large-scale network.(3)Based on the Beijing subway network, two case studies are conducted to explore the effectiveness and efficiency of the proposed method. In the first case, a simulation-based passenger flow loading model is designed with fixed capacity and strict boarding priorities (FCFS) to illustrate the accuracy of the suggested model. In the second case, the proposed model is used for the actual AFC data, and the result shows that the model has a good accuracy and computational efficiency in a large-scale subway network.

The rest of this paper is organized as follows: Section 2 gives the problem statement; Section 3 gives assumptions of this paper; Section 4 presents the detailed derivation process for the estimation of the passenger–train matching probability and the calculation method of passenger flow distribution; in Section 5 based on the synthetic data, the accuracy of the model is verified from the perspectives of passenger–train matching probability and train load, based on the real AFC data. The applicability of the model is verified by comparing it with the operator data and the control method result; Section 6 provides conclusions of our work and discusses future research directions.

## 2. Problem Description

### 2.1. Notations

In this section, the notations used in this paper are introduced, as shown in Table 1.

### 2.2. Passenger’s Trajectory

The passenger travel process in a subway network is shown in Figure 1. That is, from tap-in at the entrance gate at the origin station, then walking to the departure platform to wait for the train, then passing some stations on the train. If the passenger needs to transfer, then he/she will walk to the departure platform of the next line through the transfer channel at the transfer station and repeat the above process. When arriving at the destination station, the passenger will walk to the exit gate and tap-out to complete the subway trip. To describe the detailed travel process of a passenger more clearly in a subway network, the following concepts are defined:
Trip refers to the travel record of a passenger’s journey from the origin to the destination, including the tap-in time, tap-in station, tap-out time and tap-out station.Leg describes the movement of a passenger on a single train. As shown in Figure 1, the leg begins from the platform where the passenger boards and ends at the platform where the passenger alights. Obviously, there is at least one leg in a passenger journey.Itinerary refers to the combination of trains/services that a passenger may take on each trip. Each combination includes one or more trains/services sorted in chronological order. It is worth noting that there is only one train/service for each leg in each combination.

Without loss of generality, on a path that includes one transfer station and two legs, Figure 2 shows the possible itineraries for passenger i∈I traveling along the path. The horizontal axis is the passenger’s journey, and the vertical axis is the passenger’s travel time. The solid line is the possible trains/services on each leg; the dashed line is the passenger’s walking activity, and the dotted line is used to indicate the composition of the passenger’s journey. It can be shown that the time–space trajectory of this passenger, including his/her journey legs and the possible itineraries, can be described as follows. Passenger i enters the entry gate of the origin station at time tia and walks to the origin platform (also the departure platform of his/her first journey leg). Then passenger i may take train vi,11, vi,12 or vi,13 to reach the arrival platform of the first leg. After that, passenger i walks through a transfer channel to start his\her second leg (by taking train vi,21 or vi,22). Finally, passenger i walks to the exit gate and ends his/her journey at time tie. If the time–space trajectories of all passengers can be obtained, then by accumulating their time–space trajectories according to some rules, the subway passenger flow distribution indicators, such as the number of passengers onboard and the number of passengers on the platform, can be estimated.

### 2.3. The Passenger–Train Matching Probability

It can be seen from Figure 2 that there are multiple itineraries within the passenger’s trip $\left[t_{i}^{a},t_{i}^{e}\right]$. Obviously, which train the passenger chooses in each leg or itinerary cannot be identified. From a statistical point of view, the possibility of passengers “choosing” each potential train or itinerary, which is related to the travel elements such as walking time and waiting time corresponding to different itineraries, is different. In other words, the passenger–train matching probability can be obtained by inferring the occurrence probability of passenger travel components such as corresponding walking time and waiting time. In particular, the formulation of the matching probability requires careful consideration of the following four aspects.

#### 2.3.1. Passenger Preference

According to [14,27], passengers’ boarding preferences may vary depending on the crowding levels in trains and on platforms. For example, Figure 3 shows the diagram of the change of passengers’ willingness to take trains under different conditions. It can be seen that the possibility of passengers boarding different trains varies significantly due to the influence of factors such as the time of arrival at the platform, the number of passengers waiting at the platform, the queuing position and the crowding of arriving trains.

#### 2.3.2. Passenger Retention

Retention refers to the waiting behavior of passengers who give up the current train and wait for the next train. There are many reasons for this phenomenon, such as passenger preference, no available capacity of the current train, etc. Retention will lead to the increase of passenger travel time and the number of their possible trains, thus affecting the passenger–train matching probability.

#### 2.3.3. Interdependence of Legs

Considering that the waiting time distribution has strong time-dependent characteristics, it is necessary to calculate the probability of a passenger taking different trains in the current leg according to his/her arrival time, and a passenger’s arrival time is closely related to the access/transfer walking time and the train he/she took in previous legs. In other words, although the waiting time distribution at each platform has an independent impact on a passenger’s travel time, a passenger’s train choice behavior in different legs is interrelated; This makes the estimation of passenger–train matching probability more complicated. For example, in Figure 2, if the train that the passenger boarded in the first leg is $v_{i1}^{1}$, he/she may take train $v_{i2}^{1}$ or $v_{i2}^{2}$ on the second leg; if $v_{i1}^{2}$ is the train he/she boarded, the train on the second leg can only be $v_{i2}^{2}$.

#### 2.3.4. Heterogeneity of Passengers

Figure 4 shows the distribution of passenger travel time by taking the OD with a unique route and with multi-routes as examples, respectively. It can be seen that the passenger travel time varies greatly in both the short time slot and the long time slot dimensions. Due to the different tap-in time $t_{i}^{a}$ and travel time $t_{i}^{e}-t_{i}^{a}$ of each passenger, the possible itineraries of each passenger are also different. In other words, the passenger–train matching probability of each passenger needs to be calculated separately, which is a great challenge to the computational efficiency of the model. Therefore, improving the computational efficiency to adapt to a large-scale subway network is also one of the purposes of this study.

To sum up, the passenger’s choice behavior is mainly affected by passenger preference, retention and interdependence of journey legs, which is a complicated decision. Hence, calculating passenger–train matching probability is a complex and time-consuming problem. Therefore, the purpose of this study is to propose a probability model that can quantify the impact of these factors on a passenger’s choice behavior; Then, based on the passenger–train matching probability, the passenger flow distribution indicators such as train load are calculated. The calculation flow is shown in Figure 5 in which the passenger–train matching probability model is marked with the dashed box. First, the generation method of the passenger’s potential train set on a given path is given. Then, the passenger–train matching probability model is established based on the passenger’s travel data (record in AFC data), timetable data and the topology data of the subway network. The model quantifies the influence of walking time distribution, time-varying waiting time distribution and the dependency between trains of adjacent legs on the calculation of the passenger–train matching probability. Finally, by accumulating the passenger–train matching probability in the time and space dimension, the subway network performance indicators are obtained.

To improve the computational efficiency, this study reduces the solution scale from two aspects, which avoids many unnecessary calculations. At the path level, the dominant rules are used to identify passengers’ path choice. In terms of itinerary, we propose a passenger itinerary network decomposition method, which effectively reduces the scale of a passenger’s itineraries. As shown in Figure 2, the passenger’s itinerary network includes four potential itineraries, namely $\left\{v_{i,1}^{1},v_{i,2}^{1}\right\}$, $\left\{v_{i,1}^{1},v_{i,2}^{2}\right\}$, $\left\{v_{i,1}^{2},v_{i,2}^{1}\right\}$ $\left\{v_{i,1}^{2},v_{i,2}^{2}\right\}$. In itinerary $\left\{v_{i,1}^{2},v_{i,2}^{1}\right\}$, the transfer time and waiting time between trains $v_{i,1}^{2}$ and $v_{i,1}^{2}$ are close to 0. If the passenger could choose this itinerary: (a) the passenger instantly walks from the arrival platform of the first leg to the departure platform of the second leg, and (b) the passenger’s travel behavior in this itinerary is highly inconsistent [10,23]. The above two situations are usually considered unreasonable and rare. In fact, according to the walking time distribution and the waiting time distribution, the probability of a passenger choosing this itinerary is 0. However, if we cannot eliminate this kind of itinerary in advance in the model, it will cost about 25% extra computing power. In Section 1, we will introduce the decomposition method of the passenger itinerary network in detail.

It should be noted that the access/egress/transfer walking time distribution, waiting time distribution and passenger path choice are important inputs to the passenger–train matching probability model, which can be calculated by using the data-driven path choice inference method [21].

## 3. Assumptions

Combined with existing research, the following assumptions are made to calculate the passenger–train matching probability:

A1: A passenger’s walking time is not affected by congestion and other factors, and the walking speed is a constant personal characteristic during their trip [10,23]. This assumption is consistent with the conclusion of our field observation; that is, the walking speed of passengers has a very high consistency in the whole trip. In some time periods, measures such as inbound passenger flow control may delay passengers’ walking speed and affect the walking consistency of passengers. However, the increased walking time for this reason can be regarded as part of their waiting time at the subsequent platform [21]. Therefore, this assumption will not undermine the accuracy of the model.

A2: At the same station, the distance from different gates to the platform is the same [24].

A3: In the same time period, the waiting time of passengers on the same platform obeys the same probability distribution [13,25].

## 4. Modeling Framework

### 4.1. Feasible Train Set

Before modeling the passenger–train matching probability, we need to generate the trains set for each passenger, comprising all possible trains. Obviously, all trains within the passenger’s travel time $\left[t_{i}^{a},t_{i}^{e}\right]$ are possible alternatives. Nevertheless, given the scale of passenger volume and the complexity of the network, a rough generation method may obtain plenty of invalid possible alternatives. For example, in Figure 2, the train $v_{i1}^{3}$ is an invalid choice. When there are multiple legs in one trip, the number of alternative trains will increase by orders of magnitude, which will cause great trouble to the computational efficiency. In this case, we need to pay special attention to filtering out those irrational alternatives. As the running time of trains between any two stations on the same line is fixed, while the time the passenger spent on each leg cannot be less than the train running time, the impossible trains can be eliminated by using the train running time. Therefore, for any passenger i, assuming that their journey includes n legs, and $b_{n}$ is the operation time of the train on the leg n, his/her feasible trains need to meet the following conditions:

The departure time $o_{i,n}^{j}$ of train j from the origin platform of leg n needs to meet:(1)$$t_{i}^{a}+\overset{n-1}{\underset{n^{\prime }=1}{\sum }}b_{n^{\prime }}<o_{i,n}^{j}$$

That is, the cumulative time spent by passenger i on the n−1 legs needs to be greater than the sum of the train operation time on the corresponding legs. In other words, when passenger i arrives at platform $o_{i,n}$, it is necessary to ensure that the trains on the previous legs are feasible.

The arrival time $d_{in}^{j}$ of train j at the destination platform of leg n should be earlier than passenger $i^{\prime }$s tap-out time $t_{i}^{e}$:(2)$$d_{i,n}^{j}+\overset{N}{\underset{n^{\prime }=n+1}{\sum }}b_{n^{\prime }}<t_{i}^{e}$$

That is, when passenger i arrives at platform $d_{i,n}$, there is still enough time to complete the remaining journey.

For any two adjacent legs, if passenger i can catch train j′ of leg n after leaving from train j of the previous leg n−1, then the arrival time $d_{i,n-1}^{j}$ of train j at the destination platform of the leg n−1 must be earlier than the departure time $o_{i,n}^{j^{\prime }}$ of train j′:(3)$$o_{i,n}^{j^{\prime }}>d_{i,n-1}^{j}$$

### 4.2. The Calculation of Passenger–Train Probability

This paper proposes a reverse calculation method of passenger–train probability based on network decomposition. The core of this method is to decompose the feasible train network into multiple subnets according to the feasible train set on the last leg, where the (number of) subnets corresponds to the (number of) trains on the last leg one by one, and the train set on other legs of each subnet can be calculated according to Equations (1)–(3) and Assumption A1. Then, based on passenger trip data, path structure and the time-dependent characteristics of the network such as walking time probability distribution and waiting time probability distribution, the probability of passengers taking different trains in each subnet is calculated separately. Finally, the passenger–train probability on each leg is obtained by accumulating the passenger–train probabilities on different subnets. The method not only reduces the computational complexity, but also filters out the unreasonable itinerary and avoids many unnecessary calculations.

As in Figure 2, passenger i has two feasible trains, $v_{i,2}^{1}$ and $v_{i,2}^{2}$, on the last leg, so his/her itineraries network can be decomposed into two subnets, namely (a) and (b) in Figure 6. In the first leg, train $v_{i,1}^{1}$ appears in two subnets, respectively. It can be seen that the access time, waiting time and egress time in subnet (a) and (b) are different when passenger i takes train $v_{i,1}^{1}$, and so the probability of boarding train $v_{i,1}^{1}$ equals the sum of its conditional probability in the two subnets. The detailed derivation process of passenger–train matching probability is as follows.

Suppose passenger i has a journey with $N=\left\{1,\ldots ,n\right\}$ legs. The conditional probability of passenger i to have boarded train j on the last n given that he/she tapped-out at $t_{i}^{e}$ can be obtained by Bayes’ theorem.
(4)$$P_{i}\left(v_{i,n}^{j}|t_{i}^{e}\right)=\frac{P_{i}\left(v_{i,n}^{j},t_{i}^{e}\right)}{P_{i}\left(t_{i}^{e}\right)},\forall \;i\in I,\;j\in \;J_{i,n}$$

Using the law of total probability and the denominator of Formula (4), the probability for the passenger to tap-out at $t_{i}^{e}$ is the sum of the probabilities of boarding any feasible train on the last leg:(5)$$P_{i}\left(t_{i}^{e}\right)=\overset{J_{i,n}}{\underset{j^{\prime }=1}{\sum }}P_{i}\left(v_{i,n}^{j^{\prime }},t_{i}^{e}\right),\forall \;i\in I$$

Substituting Formula (5) into Formula (4), we can obtain:(6)$$P_{i}\left(v_{i,n}^{j}|t_{i}^{e}\right)=\frac{P_{i}\left(v_{i,n}^{j},t_{i}^{e}\right)}{\sum_{j^{\prime }=1}^{J_{i,n}}P_{i}\left(v_{i,n}^{j^{\prime }},t_{i}^{e}\right)},\forall \;i\in I,\;j\in \;J_{i,n}$$

The probability that passenger i boarded train j on leg n and tapped-out at $t_{i}^{e}$ involves three independent events: having the access/transfer time equal to $c_{i,n}^{g}$, boarding train j and having the egress time equal to $e_{i}^{j}$. Due to the differences in the facilities of the station channel (such as elevators, stair lengths, etc.), passengers’ access/transfer time and egress time can be regarded as two independent events. Similarly, affected by the number of waiting passengers, the queuing position of passengers and other factors, passengers may not be able to catch the first train after arriving on the platform, so the waiting time and walking time of passengers can also be considered to be independent of each other. Hence, the probability $P_{i}(v_{i,n}^{j},t_{i}^{e})$ is the product of the probabilities of having the access/transfer time equal to $c_{i,n}^{g}$, boarding train j and having egress time equal to $e_{i}^{j}$:(7)$$P_{i}(v_{i,n}^{j},t_{i}^{e})=P_{i}(c_{i,n}^{g})P_{i}(v_{i,n}^{j})P_{i}(e_{i}^{j}),\forall \;i\in I,\;j\in \;J_{i,n},\;g=j$$

According to the principle of network decomposition, in the last leg train j corresponds to the subnet g, so $c_{i,n}^{g}$ can be expressed as $c_{i,n}^{g}=\eta_{i,n}\times e_{i}^{j}=c_{i,n}^{j}$. Therefore, Formula (7) can be written as:(8)$$P_{i}(v_{i,n}^{j},t_{i}^{e})=P_{i}(c_{i,n}^{j})P_{i}(v_{i,n}^{j})P_{i}(e_{i}^{j}),\forall \;i\in I,\;j\in \;J_{i,n}$$

As shown in Figure 3, affected by the train passenger i boarded in the previous leg, the passenger needs a different waiting time to catch up train j on leg n. Therefore, Formula (8) can be further expressed as the sum of the joint probability of different waiting time and the other two items:(9)$$P_{i}(v_{i,n}^{j},t_{i}^{e})=P_{i}(c_{i,n}^{j})\overset{J_{i,n-1}^{g}}{\underset{k=1}{\sum }}P_{i}(w_{i,n,g}^{k,j})P_{i}(e_{i}^{j}),\forall \;i\in I,\;j\in \;J_{i,n},g=j$$

The conditional distribution for egress time can be derived based on passenger i’s feasible train set. Since passengers only have a limited number of feasible trains on the last leg (see Figure 2), the conditional probability density function of possible egress time is not continuous but discrete. Hence, the possibility of the passenger’s egress time from platform $d_{i,n}$ can be derived by discretizing the probability density function $fe_{d_{i,n}}\left(t\right)$.
(10)$$P_{i}(e_{i}^{j})=\int_{e_{i}^{j}-1}^{e_{i}^{j}}fe_{d_{i,n}}\left(t\right)dt,\forall \;i\in I,\;j\in \;J_{i,n}$$

Similarly, the probability of access/transfer time from platform $d_{i,n-1}$ to $o_{i,n}$ can be derived by discretizing the probability density function $fa_{o_{i,n}}^{d_{i,n-1}\;}\left(t\right)$ in seconds interval.
(11)$$P_{i}(c_{i,n}^{j})=\int_{c_{i}^{j}-1}^{c_{i}^{j}}fa_{o_{i,n}}^{d_{i,n-1}\;}\left(t\right)dt,\forall \;i\in I,\;j\in \;J_{i,n}$$

From Figure 3, when passenger i takes train $j\in \;J_{i,n}$ on leg n, the waiting time $w_{i,n,g}^{k,j}$ is equal to the departure time of train j minus the arrival time of train $k\in \;J_{i,n-1}$ on the previous leg n−1, and then minus the passenger’s transfer time $c_{i,n}^{g}$. Since the passenger’s transfer time is discrete, the waiting time is also discrete. Therefore, by discretizing the probability density of waiting time in seconds interval, the probability of the passenger’s waiting time can be obtained:(12)$$P_{i}(e_{i}^{j})=\int_{w_{i,n,g}^{k,j}-1}^{w_{i,n,g}^{k,j}}fw_{d_{i,n-1}^{k}}^{o_{i,n}}\left(t\right)dt,\forall \;i\in I,k\in J_{i,n-1}^{g},\;j\in J_{i,n-1}^{g},\;g\in G_{i}$$

Substituting Formulas (8)–(12) into Formula (6), the probability for passenger i boarding train j on leg n can be derived:(13)$$\begin{array}{c} P_{i}(v_{i,n}^{j}|t_{i}^{e})=\frac{P_{i}\left(c_{i,n}^{j}\right)\sum_{k=1}^{J_{i,n-1}^{g}}P_{i}\left(w_{i,n,g}^{k,j}\right)P_{i}\left(e_{i}^{j}\right)}{\sum_{j^{\prime }=1}^{J_{i,n}}\left(P_{i}\left(c_{i,n}^{j^{\prime }}\right)\sum_{k=1}^{J_{i,n-1}^{g}}P_{i}\left(w_{i,n,g}^{k,j^{\prime }}\right)P_{i}\left(e_{i}^{j^{\prime }}\right)\right)}\\ =\frac{\int_{c_{i,n}^{j}-1}^{c_{i,n}^{j}}fa_{o_{i,n}}^{d_{i,n-1}}\left(t\right)dt\sum_{k=1}^{J_{i,n-1}^{g}}\left(\int_{w_{i,n,g}^{k,j}-1}^{w_{i,n,g}^{k,j}}fw_{d_{i,n-1}^{k}}^{o_{i,n}}\left(t\right)dt\right)\int_{e_{i}^{j}-1}^{e_{i}^{j}}fe_{d_{i,n}}\left(t\right)dt}{\sum_{g=1}^{G_{i}}\sum_{j^{\prime }=1}^{J_{i,n}^{g}}\left(\int_{c_{i,n}^{j^{\prime }}-1}^{c_{i,n}^{j^{\prime }}}fa_{o_{i,n}}^{d_{i,n-1}}\left(t\right)dt\sum_{k=1}^{J_{i,n-1}^{g}}\left(\int_{w_{i,n,g}^{k,j^{\prime }}-1}^{w_{i,n,g}^{k,j^{\prime }}}fw_{d_{i,n-1}^{k}}^{o_{i,n}}\left(t\right)dt\right)\int_{e_{i}^{j^{\prime }}-1}^{e_{i}^{j^{\prime }}}fe_{d_{i,n}}\left(t\right)dt\right)},\forall \;i\in I,j\in J_{i,n}^{g},g\in G_{i} \end{array}$$

According to the passenger’s journey structure (see Figure 2) and the probability additive rule, the conditional probability of passenger i to have boarded train $k\in \;J_{i,n-1}$ on leg n−1 given that he/she tapped-out at $t_{i}^{e}$ is equal to the sum of the marginal probabilities of passenger i taking the itineraries containing train k in different subnets g:(14)$$P_{i}(v_{i,n-1}^{k}|t_{i}^{e})=\overset{G_{i}}{\underset{g=1}{\sum }}\overset{J_{i,n}^{g}}{\underset{k=1}{\sum }}P_{i}(v_{i,n-1}^{k},v_{i,n}^{j}|t_{i}^{e}),\forall \;i\in I,\;k\in \;J_{i,n-1}$$

Using Bayes’ theorem, $P_{i}(v_{i,n-1}^{k},v_{i,n}^{j},t_{i}^{e})$ can be expressed as:(15)$$P_{i}(v_{i,n-1}^{k},v_{i,n}^{j}|t_{i}^{e})=\frac{P_{i}(v_{i,n-1}^{k},v_{i,n}^{j},t_{i}^{e})}{P_{i}\left(t_{i}^{e}\right)}=\frac{P_{i}(v_{i,n-1}^{k},v_{i,n}^{j},t_{i}^{e})}{\sum_{g=1}^{G_{i}}\sum_{k^{\prime }=1}^{J_{i,n}^{g}}P_{i}(v_{i,n-1}^{k^{\prime }},v_{i,n}^{j}|t_{i}^{e})},\forall \;i\in I,\;k\in \;J_{i,n-1}$$
where $P_{i}(v_{i,n-1}^{k},v_{i,n}^{j},t_{i}^{e})$ is the joint probability of multiple independent events, namely: having the access/transfer time equal to $c_{i,n-1}^{j}$ at the origin platform of leg n−1, boarding train k on leg n−1, the transfer time is $c_{i,n}^{j}$ at the origin platform of leg n, taking train j on leg n and having egress time equal to $e_{i}^{j}$. That is:(16)$$P_{i}(v_{i,n-1}^{k},v_{i,n}^{j},t_{i}^{e})=P_{i}(c_{i,n-1}^{j})P_{i}(v_{i,n-1}^{k})P_{i}(c_{i,n}^{j})P_{i}(v_{i,n}^{j})P_{i}(e_{i}^{j}),\forall \;i\in I,\;k\in \;J_{i,n-1},\;j\in \;J_{i,n}$$

Since the probability of boarding some train is equal to the sum of the waiting time probabilities to take this train in different subnets (see Figure 3 and Figure 6), so Formula (16) can be rewritten to:(17)$$P_{i}(v_{i,n-1}^{k},v_{i,n}^{j},t_{i}^{e})=\underset{g\in G_{i}}{\sum }\left(\underset{l\in J_{i,n-2}^{g},j\in \;J_{i,n}^{g}}{\sum }P_{i}(c_{i,n}^{j})P_{i}(w_{i,n-1,g}^{l,k})P_{i}(c_{i,n}^{j})P_{i}(w_{i,n,g}^{k,j})P_{i}(e_{i}^{j})\right),\forall \;i\in I,\;k\in \;J_{i,n-1},j\in \;J_{i,n}$$

Substituting Formulas (15) and (17) into Formula (14), the probability of passenger i boarding train k on leg n−1 can be derived:(18)$$P_{i}(v_{i,n-1}^{k}|t_{i}^{e})=\frac{\sum_{g\in G_{i}}\left(\sum_{l\in J_{i,n-2}^{g},j\in \;J_{i,n}^{g}}P_{i}\left(c_{i,n}^{j}\right)P_{i}\left(w_{i,n-1,g}^{l,k}\right)P_{i}\left(c_{i,n}^{j}\right)P_{i}\left(w_{i,n,g}^{k,j}\right)P_{i}\left(e_{i}^{j}\right)\right)}{\sum_{k^{\prime }\in J_{i,n-1}}\sum_{g\in G_{i}}\left(\sum_{l\in J_{i,n-2}^{g},j\in \;J_{i,n}^{g}}P_{i}\left(c_{i,n}^{j}\right)P_{i}\left(w_{i,n-1,g}^{l,k^{\prime }}\right)P_{i}\left(c_{i,n}^{j}\right)P_{i}\left(w_{i,n,g}^{k^{\prime },j}\right)P_{i}\left(e_{i}^{j}\right)\right)},\forall \;k\in \;J_{i,n-1}$$

Finally, the general form of the probability for passenger i boarding train j on leg n−2, given that he/she tapped-out at $t_{i}^{e}$ can be derived:(19)$$\begin{array}{cc} P_{i}(v_{i,n-2}^{j}|t_{i}^{e})\\ =\frac{\sum_{g\in G_{i}}\left(P_{i}\left(c_{i,n-2}^{g}\right)\sum_{k\in \;J_{i,n-1}^{g}}P_{i}\left(w_{i,n,-2,g}^{k,j}\right)\left(\prod_{\begin{array}{c} n^{\prime }>n+1,\\ l\in J_{i,n^{\prime }-1}^{g},h\in \;J_{i,n^{\prime }}^{g} \end{array}}\left(P_{i}\left(c_{i,n^{\prime }}^{g}\right)P_{i}\left(w_{i,n^{\prime },g}^{l,h}\right)\right)\right)P_{i}\left(e_{i}^{g}\right)\right)}{\sum_{j^{\prime }\in J_{i,n-2}}\left(\sum_{g\in G_{i}}\left(P_{i}\left(c_{i,n-2}^{g}\right)\sum_{k\in \;J_{i,n-1}^{g}}P_{i}\left(w_{i,n-2,g}^{k,j^{\prime }}\right)\left(\prod_{\begin{array}{c} n^{\prime }>n+1,\\ l\in J_{i,n^{\prime }-1}^{g},h\in \;J_{i,n^{\prime }}^{g} \end{array}}\left(P_{i}\left(c_{i,n^{\prime }}^{g}\right)P_{i}\left(w_{i,n^{\prime },g}^{l,h}\right)\right)\right)P_{i}\left(e_{i}^{g}\right)\right)\right)} & ,\forall \;i\in I,\;j\\ \in \;J_{i,n-2} \end{array}$$
where the access/transfer time probability $P_{i}(c_{i,n}^{g})$ and waiting time probability $P_{i}(w_{i,n,g}^{l,h})$ can be obtained by Formulas (11) and (12), respectively.

For ease of reading, the number of legs is generalized from n−2 to n as follows:(20)$$\begin{array}{ccc} P_{i}(v_{i,n}^{j}|t_{i}^{e})= & \frac{\sum_{g\in G_{i}}\left(P_{i}\left(c_{i,n}^{g}\right)\sum_{k\in \;J_{i,n-1}^{g}}P_{i}\left(w_{i,n,g}^{k,j}\right)\left(\prod_{\begin{array}{c} n^{\prime }>n+1,\\ l\in J_{i,n^{\prime }-1}^{g},h\in \;J_{i,n^{\prime }}^{g} \end{array}}\left(P_{i}\left(c_{i,n^{\prime }}^{g}\right)P_{i}\left(w_{i,n^{\prime },g}^{l,h}\right)\right)\right)P_{i}\left(e_{i}^{g}\right)\right)}{\sum_{j^{\prime }\in J_{i,n}}\left(\sum_{g\in G_{i}}\left(P_{i}\left(c_{i,n}^{g}\right)\sum_{k\in \;J_{i,n-1}^{g}}P_{i}\left(w_{i,n,g}^{k,j^{\prime }}\right)\left(\prod_{\begin{array}{c} n^{\prime }>n+1,\\ l\in J_{i,n^{\prime }-1}^{g},h\in \;J_{i,n^{\prime }}^{g} \end{array}}\left(P_{i}\left(c_{i,n^{\prime }}^{g}\right)P_{i}\left(w_{i,n^{\prime },g}^{l,h}\right)\right)\right)P_{i}\left(e_{i}^{g}\right)\right)\right)} & ,\forall \;i\in I,\;j\\ & \;\in \;J_{i,n} \end{array}$$

When N=1, passenger i′s journey has only one leg. Formula (20) reduces to:(21)$$P_{i}(v_{i,n}^{j}|t_{i}^{e})=\frac{P_{i}\left(c_{i,n}^{j}\right)\sum_{k\in \;J_{i,n-1}^{g}}P_{i}\left(w_{i,n,j}^{k,j}\right)P_{i}\left(e_{i}^{j}\right)}{\sum_{g\in G_{i}}\sum_{\;\;j^{\prime }\in J_{i,n-1}^{g}}\left(P_{i}\left(c_{i,n}^{j^{\prime }}\right)\sum_{k\in \;J_{i,n-1}^{g}}P_{i}\left(w_{i,n,j^{\prime }}^{k,j^{\prime }}\right)P_{i}\left(e_{i}^{j^{\prime }}\right)\right)},\forall \;i\in I,\;j\in \;J_{i,n}$$

That is, Formulas (21) and (22) are equivalent to Formula (13).

### 4.3. The Calculation of Passenger Flow Distribution

As an important part of the spatial–temporal distribution of subway passenger flow, the train load is an important indicator reflecting the utilization rate of trains and the crowding of carriages. The train load is equal to the sum of the probability of all passengers taking the train. Since the train load only changes at the platform, the train load in this paper refers to the number of passengers onboard when the train leaves from the platform. The load of train s leaving platform u can be estimated recursively from the load leaving the previous platform, the accumulated probabilities of alighting at the current platform and the accumulated probabilities of boarding from the current platform. The alighting passengers include the ones who take the station as the destination and transfer to other lines at the station.
(22)$$L_{s}\left(u\right)=L_{s}\left(u-1\right)-\underset{i\in I}{\sum }\left(\underset{s=v_{i,n}^{j},d_{i,n}=u}{\sum }P(v_{i,n}^{j}|t_{i}^{e})-\underset{s=v_{i,n}^{j},o_{i,n}=u}{\sum }P(v_{i,n}^{j}|t_{i}^{e})\right),\forall \;u\in U,\;s\in \;S$$
where $L_{s}\left(u\right)$ is the load of train s when leaving platform u when u is the origin platform of the line, $L_{s}\left(u\right)=0$.

## 5. Case Study

For the purpose of model illustration and verification, we applied the proposed model on the Beijing subway network. As shown in Figure 7, the network (as of 2017) consists of 19 lines with 608 km, serving 370 stations including 56 transfer stations. It serves about 5.4 million trips on average per day. Most of the passengers use a smart card or a mobile phone to pay for the ticket, and the transactions would be recorded by the AFC systems, including the tap-in and tap-out stations and corresponding times. The data stature is shown in Table 2. As the punctuality rate of the Beijing subway exceeds 99.9% (refer to the report of the Beijing Rail Transit Operation Co., Ltd.), the planned timetable is treated as the train movement data. The network topology distance data is sorted out according to the filed survey results.

### 5.1. Experimental Design

As the real-world passengers’ travel trajectory is usually unavailable, we validate the model with synthetic data. The synthetic data is generated by a simulation-based method, so it can record the “true” travel trajectory of passengers, waiting passengers and train load and other network performance indicators of interest.

To generate the synthetic data, we use a simulation-based passenger assignment method with capacity constraints. This method takes the trimmed AFC data (only including the tap-in time, origin station and destination station), walking speed, path choices, train timetable, train capacity and network topology as inputs and outputs the passenger’s tap-out time, passenger’s train ID, train load and other network performance indicators of interest.

In the simulation method, a RUM-based (Random Utility Model) path choice model [28] is used to assign paths to passengers randomly. The train capacity adopts the standard capacity of trains on each line, and the capacity constraint coefficient is set as 1.0. Assuming that the passenger’s walking speed follows a lognormal distribution with the mean 1.17 m/s, and the standard deviation 0.35 m/s, the considered time horizon is set as 7:00~12:00, which covers both morning peak hours and off-peak hours. All OD pairs of the whole network are considered in the experiments.

Next, based on the synthetic data generated by the simulation method, we will verify the proposed model from the disaggregate level and the aggregate level, respectively. That is, we compare the probabilities of boarding the “actual” train for each synthetic passenger and compare the inference result of train load and the volume of platform left-behind passengers with the “actual” (synthetic data).

### 5.2. Comparison of Boarding the “Actual” Train

Figure 8 shows the distribution of the probabilities of boarding the “actual” train estimated by our model: Figure 8a for non-transfer trips, Figure 8b–d for the trips with 1/2/multiple transfer times. The horizontal ordinate is the probability value, and the closer the value is to one, the more consistent the inference result is with the actual train. The primary ordinate is the frequency of the actual train selected by passengers with corresponding probability, and the secondary ordinate is the percentage corresponding to the frequency. It can be seen that the model has a very high accuracy (>90%) for all trip types, and the less the transfer times, the higher the accuracy of the model.

### 5.3. Comparison of Passenger Flow

The passenger flow intensity and passenger flow type are the main factors affecting a passenger’s travel decision. Generally, the passenger flow can be divided into three categories, which is the new tap-in passengers, the transfer-in passengers and the on-board passengers. To illustrate the accuracy of this model under different passenger flow types and passenger flow intensities, we conducted three comparative experiments. As shown in Figure 7, TianTongYuan (TTY) is the second platform in the downward direction of Line 5, hence the train load is only affected by the new tap-in passenger flow. LiShuiQiao (LSQ) is a transfer station between Line 13 and Line 5; the waiting passengers at this platform include the new tap-in passenger flow and the transfer-in passenger flow from Line 13. When the downward train runs to HuiXinxijieBeiKou (HXBK), the middle station of Line 5, the train is nearly full, especially during peak hours. Therefore, the remaining capacity of the arrival train is the main factor affecting the passengers at HXBK.

Figure 9 shows the train load and the volume of platform left-behind passengers for TTY(a) and LSQ(b) by our model. The horizontal ordinate is the train departure time, and the ordinate is the train load and the volume of left-behind passengers. The trend of the curves suggests that the calculated results are in good agreement with the synthetic data. In other words, our model is able to replicate the train load and the volume of platform left-behind passengers accurately.

Different from the TTY and LSQ, when the train arrives at the HXBK, the train load is heavy. In Figure 10, the synthetic data shows that there are many left-behind passengers on the platform during 7:35~8:52, and the train load also reaches the capacity constraint. Long left-behind time will make it more challenging to estimate passengers’ “actual” train. Specifically, the longer the travel time, the greater the number of potentially feasible trains, and the more difficult it is to accurately estimate the passenger’s “actual” train. The calculation results in Figure 10 show that the replications of both train load and the left-behind passenger volume are ideal. The error between the calculated train load and the “actual” train load (synthetic data) is calculated as follows: $\varepsilon =|L_{s}\left(u\right)-{\tilde{L}}_{s}\left(u\right)|/{\tilde{L}}_{s}\left(u\right)$, where s is the train ID, u is the platform ID, $L_{s}\left(u\right)$ is the calculated load of train s by the proposed model, and ${\tilde{L}}_{s}\left(u\right)$ is the “actual” train load. We can see that all the errors are less than 5%. These suggest that the proposed model can work well under large passenger flow.

To understand the results more comprehensively, Figure 11 shows the distribution of errors between the calculated result and the “actual” train load during 7:00~12:00. Since the model accuracy is high when the train load rate is low, Figure 11 only shows the statistical results of errors when the train load rate exceeds 50%. The horizontal ordinate is the error interval, the ordinate is the frequency, and the red curve is the cumulative proportion. We can see that about 90% of the errors are less than 10%, and only a few errors exceed 15%. The statistical results suggest that the calculation results are consistent with the “true” train load in most cases. Since the expected load of the train can be estimated as the sum of the probabilities of boarding the corresponding train, it can be considered that the inference results at the aggregate level are consistent with the conclusions at the disaggregate level. Therefore, we can conclude that our model can accurately reproduce the spatio–temporal distribution of passenger flow on the individual trains.

### 5.4. Practical Application for Beijing Subway Network

In this section, we apply the proposed model on the Beijing subway network with the real AFC data. The data were collected from a workday during the day in October 2017, with a total of 5,393,777 transaction records. The calculated results of the proposed model are based on two sets of data. Data-I are the reference data, which are the average hourly train load provided by the operator. Data-II are the comparative data, which are the simulation results under different train capacity constraints (CC). According to the annual report of the Beijing subway in 2017, the maximum load rate of the Beijing subway is 1.43, so the range of train capacity constraint in the simulation method is [1.0, 1.5]. Our model is implemented in Java and runs on a lab computer with a 3.8 GHz Intel Pentium 5500 processor and a RAM of 16 GB, running Windows 10. The computational times take about 82 min.

Figure 12 shows the train load rate at HXBK. According to Data-I, the average train load rate is 1.18 during 7:45~8:45(interested time period). The calculated result by the proposed model is 1.09, and the error between the result and Data-I is 7.6%, which is an acceptable accuracy in practice. More importantly, this model can reveal the variation law of train load rate from the individual level. From Figure 12, we can find that the train load rate during interested time period does not always maintain at 1.18 but fluctuates. The maximum train load appears during 7:50~8:15, which is in line with our actual travel experience.

For comparison, Figure 12 also provides the simulated results of the train load rate under different train capacity constraints. It can be seen that the train load rate is constant during the whole interested time period when the CC is less than 1.1. When it is increased to 1.2, this phenomenon is significantly alleviated, but this problem still exists for the trains during 7:50~8:15. When the CC is 1.3, the simulation results (individual train load rate) are approximate to the results of the proposed model. Table 3 gives the comparative statistical data of the two groups of calculated results. It can be seen that when the CC equals 1.1, the simulation result (average load rate) is closest to 1.18 (Data-I), but most of the train load rates are equal to CC. It illustrates that the simulation method cannot produce accurate results at the non-aggregate level and the aggregate level at the same time, while our model can give more accurate results at both the statistical and individual levels.

According to Data-I, Table 4 lists the calculation results of train load of other stations (see in Figure 7). The time period is determined by Data I. The optimal capacity constraint (OCC) refers to the capacity constraint value adopted by the simulation method when the average train load rate calculated by the simulation method is closest to the Data-I. We can obtain OCC by Formula (23):(23)$$OCC_{u}=arg\min\left\{\left|{\bar{L}}_{u}-{\bar{L}}_{c,u}\right|\right\},\;c=\left\{1.0,1.1,1.2,1.3,1.4,1.5\right\},u\in U$$
where u is the platform provided by Data-I, ${\bar{L}}_{u}$ is the average train load of the interested time period at platform u in Data-I, and ${\bar{L}}_{c,u}$ is the simulation result of platform u when CC is c.

We find that: (1) our model performs well, most of the errors are less than 10%, which is an acceptable result in practice; and (2) the OCC of the simulation method varies from line/station to line/station. In other words, the accuracy of the simulation results depends on the capacity constraint parameters. Unfortunately, the train load varies because of the change in crowding levels over time [5,13]. Therefore, even at the aggregate level, there is hardly a unified capacity constraint value that can accurately describe the average train load on different lines/stations. In comparison, our model can obtain the train load similar to the real value and does not need the assumption of capacity constraints. Therefore, our model is an effective means to estimate the subway performance indicators.

In addition, in terms of computational efficiency, although existing studies [13,23,26] have pointed out that the computational efficiency of probability-based model limits its application in a large-scale subway network, only study [13] gives some indicators related to computational efficiency. Due to the different test data, the computational efficiency of our model cannot be directly compared with the literature [13], but it can be indirectly compared by the key indexes affecting the computational time, such as the number of stations, the number of transfer stations, the number of candidate paths, and the scale of AFC data, as shown in Table 5. It can be seen that the computational time of our model is an acceptable 82 min, which is significantly better than the existing research when the network indexes are approximate.

## 6. Conclusions

Passenger flow distribution is the basis for operators to formulate and adjust service plans. High accuracy time–space distribution information of passenger flow is particularly important for providing efficient and high-quality services in a densely used large subway network. In addition, knowing the position and number of passengers at a given time is also a key issue in case of disruption/emergency, which is very helpful for operators to provide a quick response such as introducing shuttle bus services [2]. Therefore, an effective and easily implementable passenger flow distribution model is appealing.

Therefore, based on the AFC data, train timetables and subway network topology, this paper proposes a method to calculate subway passenger flow distribution based on passenger–train matching probability. To calculate the passenger–train matching probability, first, the complex dependence between passenger travel time, departure time, travel path structure and path travel time is modeled. Then, aiming at this model, a method of reversely deriving the probability of passengers taking different trains is proposed. This method can decompose the network composed of the potential trains that passenger may board into multiple sub-networks from the structure, which effectively reduces the scale of the feasible train combination set and improves the calculation efficiency. To verify the accuracy and applicability of the model, we first applied the model to the synthetic data. The results show that the model has a high accuracy in estimating the “real” train passengers boarding and depicting the change of train load. Then, we apply the model to real AFC data with more noise and greater uncertainty. The results show that compared with the existing research our model has significant advantages in computational efficiency when the number of passenger trips, network size and other key indicators are close.

Although the model has shown good problem-solving ability, there are still some limitations: (1) This paper assumes that the distance from the platform to the gate is the same at the same station. Considering that the platform length cannot be ignored, this assumption may affect the model accuracy to some extent. (2) The access/transfer/egress time distribution are obtained from the raw AFC data, which are sensitive to data quality. Refining the location of the passenger fare gate and simplifying the pedestrian passage in the passenger station is conducive to improving the regularity of passenger walking time distribution. Nevertheless, the proposed model has demonstrated a good capacity to solve the problem in terms of accuracy and efficiency. The passenger train choice is the core of subway passenger flow distribution. Therefore, the model can also be extended in many directions, such as estimating the number of passengers left-behind on the platform, so as to help operators better observe the performance of the subway system.

In future, we can carry out or extend our research from the following aspects: (1) attempting to collect the complete travel chain of passengers anonymously through advanced technologies such as wearable devices to verify the accuracy of our model with real data; (2) attempting to integrate more data such as Bluetooth [29], Wi-Fi [30] and camera [31] to study the passenger flow distribution; (3) readjusting the use of the proposed method in other fields, such as street network traffic modeling with different constraints (such as traffic light timing); and (4) further exploring the time-dependent passenger volume, time-dependent network characteristics and more real network (such as the distribution of passengers’ position on the platform), and developing this model to facilitate the modeling of the passenger flow distribution under more complex working conditions.

## Figures and Tables

**Figure 1 entropy-24-01026-f001:**
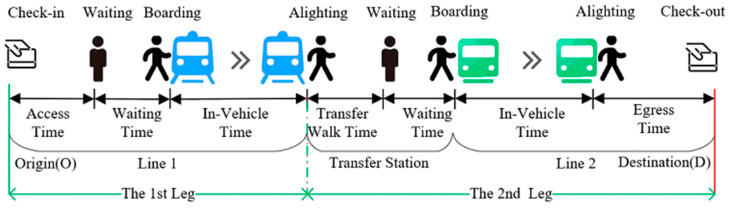
Passenger travel diagram by subway.

**Figure 2 entropy-24-01026-f002:**
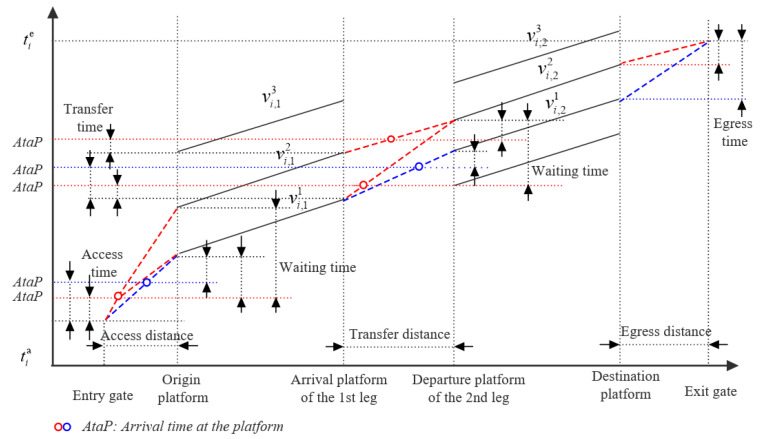
The Network of a Passenger’s Possible Trains/Services.

**Figure 3 entropy-24-01026-f003:**
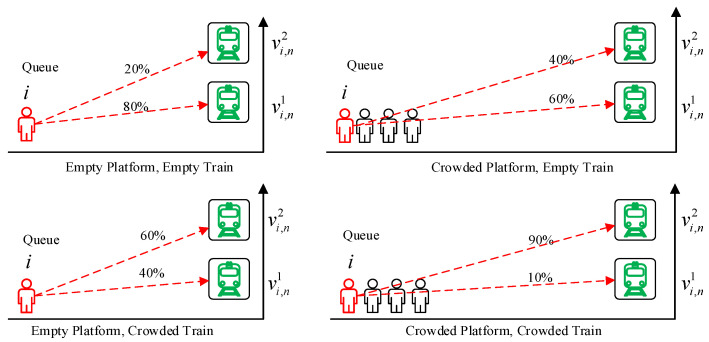
Diagram of passenger’s boarding willingness under different conditions.

**Figure 4 entropy-24-01026-f004:**
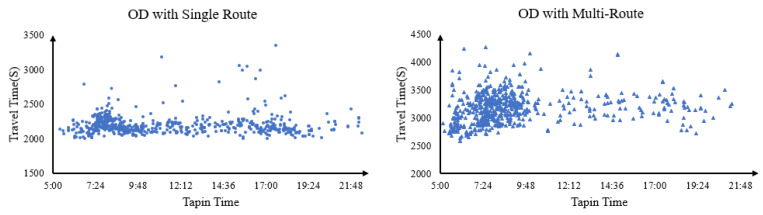
Passenger Travel Time Distribution.

**Figure 5 entropy-24-01026-f005:**
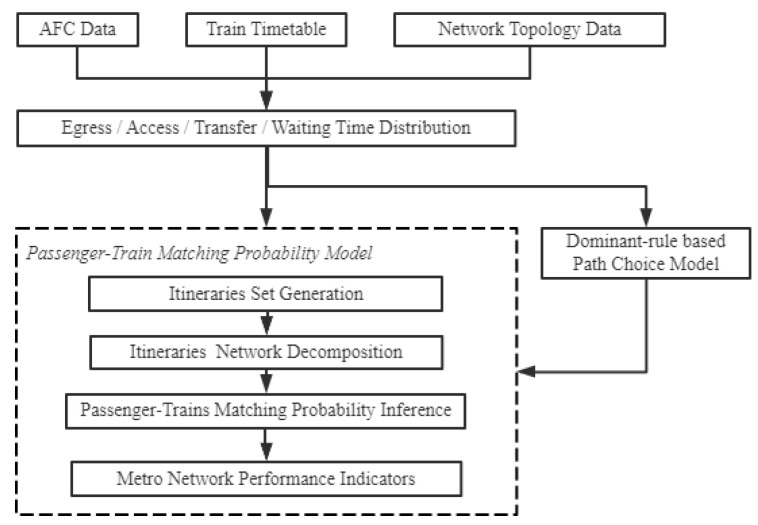
The passenger flow distribution calculation flow.

**Figure 6 entropy-24-01026-f006:**
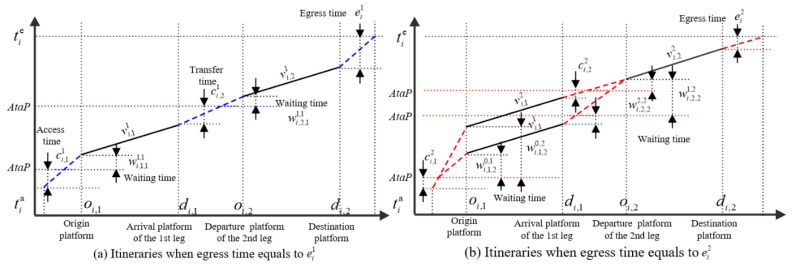
Subnets of passenger’s possible trains.

**Figure 7 entropy-24-01026-f007:**
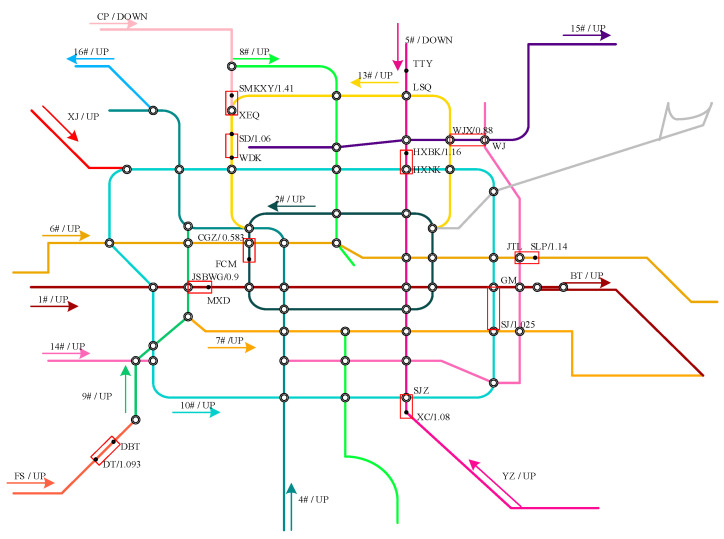
Beijing subway network.

**Figure 8 entropy-24-01026-f008:**
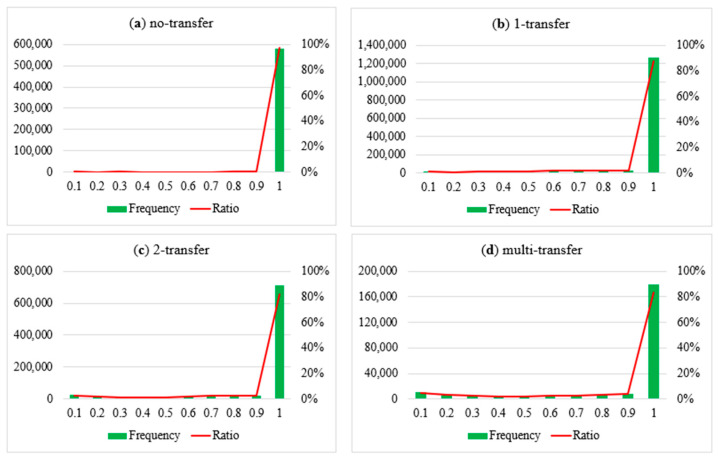
The probability distribution of boarding the “actual” train.

**Figure 9 entropy-24-01026-f009:**
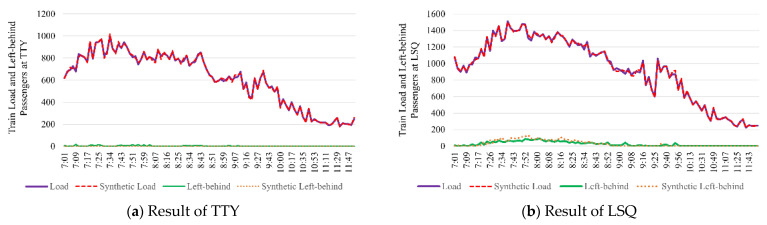
Train load and left-behind passengers at TTY/LSQ.

**Figure 10 entropy-24-01026-f010:**
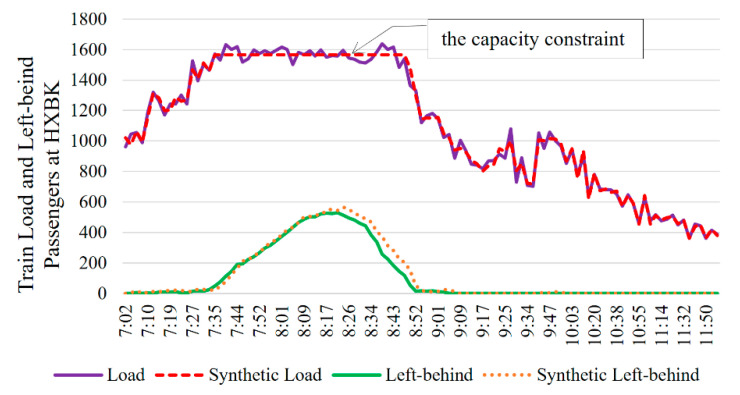
Train load and left-behind passengers at HXBK.

**Figure 11 entropy-24-01026-f011:**
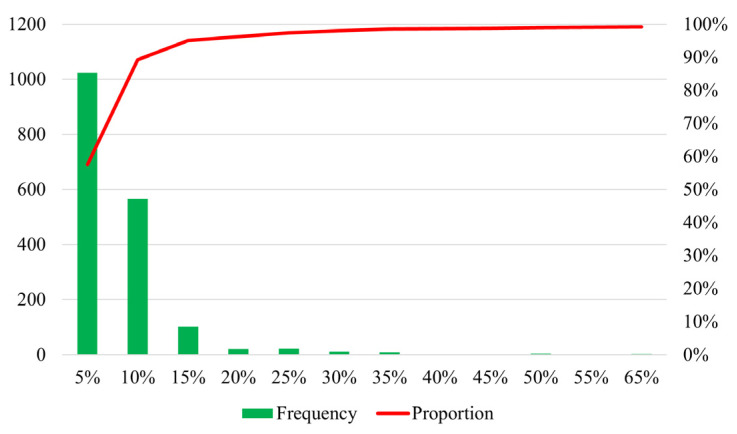
The distribution of errors between the calculated results and the “actual” train load.

**Figure 12 entropy-24-01026-f012:**
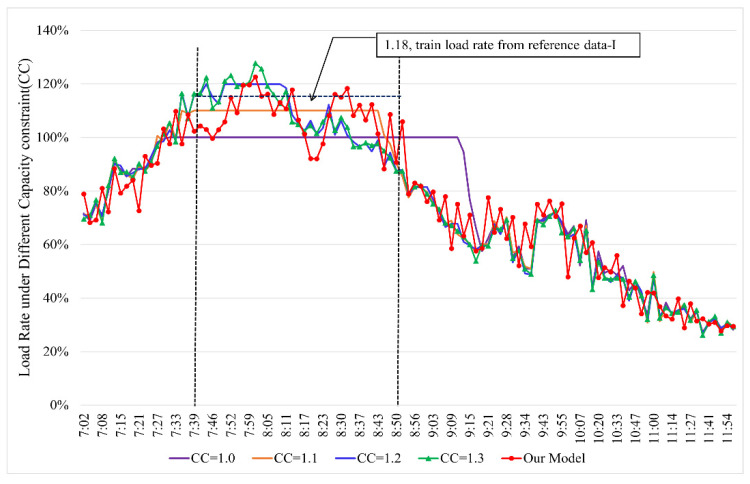
Train load at XHBK with real AFC.

**Table 1 entropy-24-01026-t001:** Notations of this paper.

Set	Description
I	the set of passengers, i ∈ I.
S	the set of train/service id in the train timetable, s∈S;
U	the set of platforms, u∈U;
$J_{i}$	the set of train, $j\in J_{i}$;
$N_{i}$	the leg number of passenger *i*’s journey, $n\in N_{i}$;
$v_{i,n}^{j}$	the train *j* of passenger *i* on the leg *n* of his/her journey, the variable as a whole corresponds to the train id in the timetable;
g	the train’s subnet index, $g\in G_{i}$;
$t_{i}^{a}$	the tap-in time of passenger *i*;
$t_{i}^{e}$	the tap-out time of passenger *i*;
$o_{i,n}$	the origin/departure platform of passenger *i* on leg *n*;
$d_{i,n}$	the destination/arrival platform of passenger *i* on leg *n*;
$o_{i,n}^{j}$	the departure time of train $v_{in}^{j}$ from the origin platform;
$d_{i,n}^{j}$	the arrival time of train $v_{in}^{j}$ to the destination platform;
$e_{i}^{j}$	the egress time of passenger *i* in the subnet *g*;
$\eta_{i,n}$	the ratio of access distance (if n=1) or transfer distance (if n>1) to egress distance $\eta_{in}>0$;
$c_{i,n}^{g}$	the transfer time of passenger *i* between the arrival platform of leg n−1 and the departure platform of leg *n* in the subnet *g*, $c_{i,n}^{g}=\eta_{i,n}\times e_{i}^{g}$$;\;\mathrm{when}\;n=1,\;\;c_{i,n}^{g}$ is the access time;
$w_{i,n,g}^{k,j}$	the waiting time of passenger *i* at the departure platform of leg *n* in subnet *g* when the passenger boards train *k* on leg n−1 and boards train *j* on leg *n*. It equals the departure time $o_{i,n}^{j}$ of train $v_{i,n}^{j}$ minus the arrival time $d_{i,n-1}^{k}$ of train $v_{i,n-1}^{k}$, and then minus the transfer time $c_{i,n}^{g}$ between the two legs, that is $w_{i,n,g}^{k,j}=o_{i,n}^{j}-d_{i,n-1}^{k}-c_{i,n}^{g}$$,\;\mathrm{and}\;w_{i,n,g}^{k,j}>0$;
$J_{i,n}$	the train set of passenger *i* on the leg *n* of his/her journey
$J_{i,n}^{g}$	the train set of passenger *i* on leg *n* of subnet *g*, $J_{i,n}=\sum_{g^{\prime }=1}^{G_{i}}J_{i,n}^{g^{\prime }}$
$fe_{u}\left(t\right)$	the egress time probability density of platform *u*;
$fw_{t}^{u}\left(t\right)$	the waiting time probability density of platform *u* at time *t*;
$fa_{u}^{u^{\prime }}\left(t\right)$	the transfer time probability density from platform $u^{\prime }$ to platform *u*, when $u^{\prime }=u$, it is the access time probability density.

**Table 2 entropy-24-01026-t002:** AFC transaction information.

ID	Origin Station	Destination Station	Tap-In Time	Tap-Out Time
20036058711	TTY	SYJ	17 October 2017 08:22:00	17 October 2017 08:57:18
…	…	…	…	…

**Table 3 entropy-24-01026-t003:** Comparison of average train load rate at HXBK.

	**Our Model**	**Simulation Method**				**Data-I**
Capacity Constraint		1.0	1.1	1.2	1.3	
Train Load Rate in Average	1.16	1.0	1.096	1.09	1.09	1.18
Number/Percentage of Train Load Rate Reaches the CC	-	28/100%	25/89%	10/36%	0/0%	-

**Table 4 entropy-24-01026-t004:** Comparison of average train load rate at different stations.

Line	1	2	6	10	13	15	CP	FS	YZ
Period	7:35~8:35	7:50~8:50	7:35~8:35	7:50~8:50	7:35~8:35	8:25~9:25	7:45~8:45	7:20~8:20	7:20~8:20
Direction/Station	UP/JSBWG	UP/CGZ	DN/SLP	UP/SJ	UP/SD	UP/WJX	DOWN/SMKXY	UP/DT	UP/XC
Data-I	0.95	0.62	1.21	0.95	1.1	0.88	1.24	1.1	1.14
Our Model/Errors	0.9	0.583	1.14	1.025	1.061	0.88	1.41	1.093	1.08
	5.14%	6.03%	5.47%	7.89%	3.55%	0%	13.71%	0.64%	5.26%
OCC	1.2	1.4	1.3	1.0	1.5	1.1	1.2	1.1	1.2

**Table 5 entropy-24-01026-t005:** Comparison of network indexes affecting computational efficiency.

	MTR	Beijing Subway
Lines	11 ^1^	19
Stations	154 ^1^	370
Transfer stations	20 ^1^	56
AFC data	5 M~7 M ^2^	5.4 M
Candidate paths	2 (Maximum) ^2^	4 (Maximum), 2.45 (In average)
PC	3.40 GHz CPU, 16 GB RAM ^2^	3.80 GHZ CPU, 16 G RAM
With Parallel computing	Unknown	No
Computational time	About 2880 min ^2^	About 82 min

^1^ From MTR official website, 2020. ^2^ From literature [13].

## Data Availability

The data presented in this study are available on request from the corresponding author.

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
