# Peer review of "A Calculation Method of Passenger Flow Distribution in Large-Scale Subway Network Based on Passenger–Train Matching Probability"

_entropy, 2022, doi:10.3390/e24081026_

Round 1
Reviewer 1 Report
The authors calculated the probability of the passenger-train matching in a subway network. Space-time incorporates temporal information and analytical functions so that (GIS) technology could handle both spatial and temporal data. This concept is well established in GIS domain. I would suggest authors to revisit other applications and debate whether the proposed model could contribute to other applications in one way or others. For example in traffic modelling on the street network with different constraints such as traffic light timing, how the proposed approach could be repurposed?
The accuracy of the results was assessed based on the synthetic data. What would be the possibility to accurately calculate the models precision. I would like to hear authors opinion if there are anyway to obtain the full path (trip) by some passenger?
Could it be possible to incorporate other source of data such as cameras, bluetooth, etc to accurately model the probability in the subway network. ? and How? What would be the challenges?
Reviewer 2 Report
It is recommended that the main findings be included in the summary.
Reinforce the conclusions
Indicate future research related to the topic investigated
Reinforce the bibliography with authors of years such as 2019, 2020, 2021.
Have a nice week
Author Response
Dear Revierer,
Please see the attachment.
Thank you.
